# A Foundation Model
# for Zero-shot Logical Query Reasoning

**Mikhail Galkin[1], Jincheng Zhou[2],[*] Bruno Ribeiro[2], Jian Tang[3,4,5], Zhaocheng Zhu[3,6]**
[1]Intel AI Lab, [2]Purdue University, [3]Mila - Québec AI Institute,
[4]HEC Montréal, [5]CIFAR AI Chair [6]Université de Montréal

## Abstract

Complex logical query answering (CLQA) in knowledge graphs (KGs) goes beyond simple KG completion and aims at answering compositional queries comprised of multiple projections and logical operations. Existing CLQA methods that learn parameters bound to certain entity or relation vocabularies can only be applied to the graph they are trained on which requires substantial training time before being deployed on a new graph. Here we present ULTRAQUERY, the first foundation model for inductive reasoning that can zero-shot answer logical queries on *any* KG. The core idea of ULTRAQUERY is to derive both projections and logical operations as vocabulary-independent functions which generalize to new entities and relations in any KG. With the projection operation initialized from a pre-trained inductive KG completion model, ULTRAQUERY can solve CLQA on any KG after finetuning on a single dataset. Experimenting on 23 datasets, ULTRAQUERY in the zero-shot inference mode shows competitive or better query answering performance than best available baselines and sets a new state of the art on 15 of them.

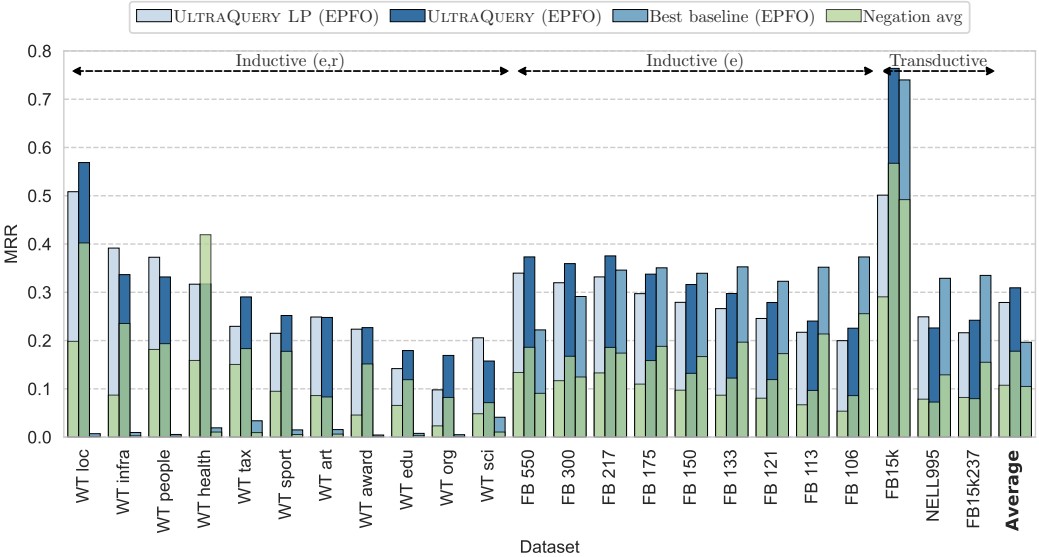

Figure 1: Zero-shot query answering performance (MRR, higher is better) of a single ULTRAQUERY model trained on one FB15k237 queries dataset compared to the best available baselines and ablated ULTRAQUERY LP on 23 datasets. *EPFO* is the average of 9 query types with $(\wedge, \vee)$ operators, *Negation* is the average of 5 query types with the negation operator $(\neg)$. On average, a single ULTRAQUERY model outperforms the best baselines trained specifically on each dataset. More results are presented in Table 2 and Appendix C.

---

[*]Work done during the internship at Intel. Code: https://github.com/DeepGraphLearning/ULTRA

38th Conference on Neural Information Processing Systems (NeurIPS 2024).

# 1 Introduction

Complex logical query answering (CLQA) generalizes simple knowledge graph (KG) completion to more complex, compositional queries with logical operators such as intersection ($\land$), union ($\lor$), and negation ($\neg$). Such queries are expressed in a subset of first-order logic (FOL) where existentially quantified ($\exists$) *variables* and given *constants* comprise *relation projections* (or *atoms*), and logical operators combine projections into a logical query (graph pattern). A typical example of a logical query [27] is presented in Figure 2: $?U.\exists V :$ Win(NobelPrize, $V$) $\land$ Citizen(USA, $V$) $\land$ Graduate($V, U$) where Win() is a relation projection, NobelPrize is a constant, and $V$ is an existentially quantified variable.

Due to the incompleteness of most KGs, these logical queries cannot be directly solved by graph traversal algorithms. Consequently, CLQA methods have to deal with missing edges when modeling the projection operators. The vast majority of existing CLQA methods [27, 26, 25, 3, 5] predict missing edges by learning graph-specific entity and relation embeddings making such approaches transductive and not

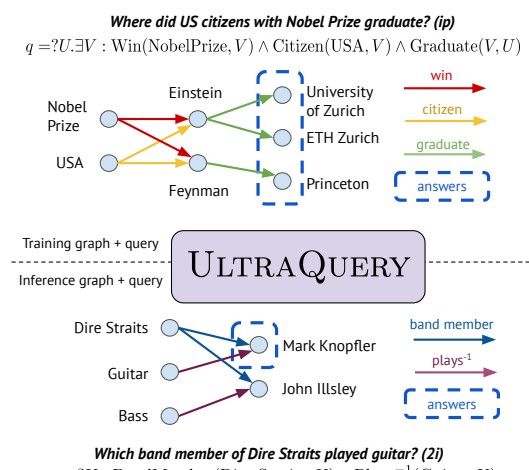

*Where did US citizens with Nobel Prize graduate? (ip)*
$q = ?U.\exists V :$ Win(NobelPrize, $V$) $\land$ Citizen(USA, $V$) $\land$ Graduate($V, U$)

*Which band member of Dire Straits played guitar? (2i)*
$q = ?U :$ BandMember(Dire Straits, $U$) $\land$ Plays$^{-1}$(Guitar, $U$)

Figure 2: The inductive logical query answering setup where training and inference graphs (and queries) have different entity and relation vocabularies. We propose a single model (ULTRAQUERY) that zero-shot generalizes to query answering on any graph with new entity or relation vocabulary at inference time.

generalizable to other KGs. A few approaches [43, 14, 17] are able to generalize query answering to new nodes at inference time but still need a fixed relation vocabulary.

In this work, we focus on the hardest inductive generalization setup where queries and underlying graphs at inference time are completely different from the training graph, *i.e.*, both entities and relations are new. Furthermore, we aim at performing CLQA in the *zero-shot* setting with one single model. That is, instead of finetuning a model on each target dataset, we seek to design a unified approach that generalizes to any KG and query at inference time. For example, in Figure 2, the training graph describes academic entities with relations Win, Citizen, Graduate[2] whereas the inference graph describes music entities with relations Band Member and Plays. The query against the inference graph $?U :$ BandMember(Dire Straits, $U$) $\land$ Plays$^{-1}$(Guitar, $U$) involves both new entities and relations and, to the best of our knowledge, cannot be tackled by any existing CLQA method that learns a fixed set of entities or relation embeddings from the training graph.

**Contributions.** Our contributions are two-fold. First, none of the existing CLQA methods can generalize to query answering over new arbitrary KGs with new entities and relations at inference time. We bridge this gap by leveraging the recent progress in inductive KG reasoning [15, 16] and devise ULTRAQUERY, the first foundation model for CLQA that generalizes to logical queries on any arbitrary KG with any entity and relation vocabulary in the zero-shot fashion without relying on any external node or edge features. ULTRAQUERY parameterizes the projection operator by an inductive graph neural network (GNN) and implements non-parametric logical operators with fuzzy logics [31]. The pre-trained projection operator [15] does not learn any graph-specific entity nor relation embeddings thanks to the generalizable meta-graph representation of relation interactions, and therefore enables zero-shot generalization to any KG.

Second, in the absence of existing datasets for our inductive generalization setup, we curate a novel suite of 11 inductive query answering datasets where graphs and queries at inference time have new entity and relation vocabularies. Experimentally, we train a single ULTRAQUERY model on one dataset and probe on other 22 transductive and inductive datasets. Averaged across the datasets, a single ULTRAQUERY model outperforms by 50% (relative MRR) the best reported baselines in the literature (often tailored to specific graphs) on both EPFO queries and queries with negation.

---

[2] We assume the presence of respective inverse relations $r^{-1}$.

## 2   Related Work

**Complex Logical Query Answering.** To the best of our knowledge, there is no existing approach for generalizable and inductive query answering where the method is required to deal with new entitis and relations at inference time.

Due to the necessity of learning entity and relation embeddings, the vast majority of existing methods like GQE [19], BetaE [25], ConE [40], MPQE [10] (and many more from the survey by Ren et al. [27]) are transductive-only and tailored for a specific set of entities and relations. Among them, CQD [3] and QTO [5] are inference-only query answering engines that execute logical operators with non-parametric fuzzy logic operators (*e.g.*, product logic) but still require pre-trained entity and relation embedding matrices to execute relation projections (link prediction steps). We refer the interested reader to the comprehensive survey by Ren et al. [27] that covers query answering theory, a taxonomy of approaches, datasets, and open challenges.

A few models [43, 14, 17] generalize only to new entities by modeling entities as a function of relation embeddings. Gebhart et al. [17] apply the idea of cellular sheaves and harmonic extension to translation-based embedding models to answer conjunctive queries (without unions and negations). NodePiece-QE [14] trains an inductive entity encoder (based on the fixed vocabulary of relations) that is able to reconstruct entity embeddings of the new graph and then apply non-parametric engines like CQD to answer

Table 1: Comparison with existing CLQA approaches. *Ind.* denotes inductive generalization to new entities *(e)* and relations *(r)*. ULTRAQUERY is the first inductive method the generalizes to queries over new entities and relations at inference time.

| Method | Ind. $e$ | Ind. $r$ | Ind. Logical Ops |
|---|---|---|---|
| Query2Box [26], BetaE [25] | ✗ | ✗ | Parametric, ✗ |
| CQD [3], FuzzQE [9], QTO [5] | ✗ | ✗ | Fuzzy, ✓ |
| GNN-QE [43], NodePiece-QE [14] | ✓ | ✗ | Fuzzy, ✓ |
| ULTRAQUERY **(this work)** | ✓ | ✓ | Fuzzy, ✓ |

queries against new entities. The most effective inductive (entity) approach is GNN-QE [43, 14] that parameterizes each entity as a function of the relational structure between the query constants and the entity itself. However, all these works rely on a fixed relation vocabulary and cannot generalize to KGs with new relations at test time. In contrast, our model uses inductive relation projection and inductive logical operations that enable zero-shot generalization to any new KG with any entity and relation vocabulary without any specific training.

**Inductive Knowledge Graph Completion.** In CLQA, KG completion methods execute the projection operator and are mainly responsible for predicting missing links in incomplete graphs during query execution. Inductive KG completion is usually categorized [8] into two branches: (i) inductive entity (inductive $(e)$) approaches have a fixed set of relations and only generalize to new entities, for example, to different subgraphs of one larger KG with one set of relations; and (ii) inductive entity and relation (inductive $(e, r)$) approaches that do not rely on any fixed set of entities and relations and generalize to any new KG with arbitrary new sets of entities and relations.

Up until recently, the majority of existing approaches belonged to the inductive $(e)$ family (*e.g.*, GraIL [29], NBFNet [42], RED-GNN [38], NodePiece [13], A*Net [44], AdaProp [39]) that generalizes only to new entities as their featurization strategies are based on learnable relation embeddings.

Recently, the more generalizable inductive $(e, r)$ family started getting more attention, *e.g.*, with RMPI [18], InGram [22], ULTRA [15], and the theory of *double equivariance* introduced by Gao et al. [16] followed by ISDEA and MTDEA [41] models. In this work, we employ ULTRA to obtain transferable graph representations and execute the projection operator with link prediction over any arbitrary KG without input features. Extending our model with additional input features is possible (although deriving a single fixed-width model for graphs with arbitrary input space is highly non-trivial) and we leave it for future work.

## 3   Preliminaries and Problem Definition

We introduce the basic concepts pertaining to logical query answering and KGs largely following the existing literature [14, 27, 15].

**Knowledge Graphs and Inductive Setup.** Given a finite set of entities $\mathcal{V}$ (nodes), a finite set of relations $\mathcal{R}$ (edge types), and a set of triples (edges) $\mathcal{E} = (\mathcal{V} \times \mathcal{R} \times \mathcal{V})$, a knowledge graph $\mathcal{G}$ is a tuple $\mathcal{G} = (\mathcal{V}, \mathcal{R}, \mathcal{E})$. The simplest *transductive* setup dictates that the graph at training time

$\mathcal{G}_{train} = (\mathcal{V}_{train}, \mathcal{R}_{train}, \mathcal{E}_{train})$ and the graph at inference (validation or test) time $\mathcal{G}_{inf} = (\mathcal{V}_{inf}, \mathcal{R}_{inf}, \mathcal{E}_{inf})$ are the same, *i.e.*, $\mathcal{G}_{train} = \mathcal{G}_{inf}$. By default, we assume that the inference graph $\mathcal{G}_{inf}$ is an incomplete part of a larger, non observable graph $\hat{\mathcal{G}}_{inf}$ with missing triples to be predicted at inference time. In the *inductive* setup, in the general case, the training and inference graphs are different, $\mathcal{G}_{train} \neq \mathcal{G}_{inf}$. In the easier inductive entity (*inductive* $(e)$) setup tackled by most of the KG completion literature, the relation set $\mathcal{R}$ is fixed and shared between training and inference graphs, *i.e.*, $\mathcal{G}_{train} = (\mathcal{V}_{train}, \mathcal{R}, \mathcal{E}_{train})$ and $\mathcal{G}_{inf} = (\mathcal{V}_{inf}, \mathcal{R}, \mathcal{E}_{inf})$. The inference graph can be an extension of the training graph if $\mathcal{V}_{train} \subseteq \mathcal{V}_{inf}$ or be a separate disjoint graph (with the same set of relations) if $\mathcal{V}_{train} \cap \mathcal{V}_{inf} = \varnothing$. In CLQA, the former setup with the extended training graph at inference is tackled by InductiveQE approaches [14].

In the hardest inductive entity and relation (inductive $(e, r)$) case, both entities and relations sets are different, *i.e.*, $\mathcal{V}_{train} \cap \mathcal{V}_{inf} = \varnothing$ and $\mathcal{R}_{train} \cap \mathcal{R}_{inf} = \varnothing$. In CLQA, there is no existing approach tackling this case and our proposed ULTRAQUERY is the first one to do so.

**First-Order Logic Queries.** Applied to KGs, a first-order logic (FOL) query $q$ is a formula that consists of constants Con (Con $\subseteq \mathcal{V}$), variables Var (Var $\subseteq \mathcal{V}$, existentially quantified), relation *projections* $R(a, b)$ denoting a binary function over constants or variables, and logic symbols ($\exists, \wedge, \vee, \neg$). The answers $A_\mathcal{G}(q)$ to the query $q$ are assignments of variables in a formula such that the instantiated query formula is a subgraph of the complete, non observable graph $\hat{\mathcal{G}}$. Answers are denoted as *easy* if they are reachable by graph traversal over the incomplete graph $\mathcal{G}$ and denoted as *hard* if at least one edge from the non observable, complete graph $\hat{\mathcal{G}}$ has to be predicted during query execution.

For example, in Figure 2, a query *Which band member of Dire Straits played guitar?* is expressed in the logical form as $?U : \text{BandMember}(\text{Dire Straits}, U) \wedge \text{Plays}^{-1}(\text{Guitar}, U)$ as an *intersection* query. Here, $U$ is a projected target variable, Dire Straits and Guitar are constants, BandMember and Plays are *relation projections* where Plays$^{-1}$ denotes the inverse of the relation Plays. The task of CLQA is to predict bindings (mappings between entities and variables) of the target variable, *e.g.*, for the example query the answer set is a single entity $\mathcal{A}_q = \{(U, \text{Mark Knopfler})\}$ and we treat this answer as an *easy* answer as it is reachable by traversing the edges of the given graph. In practice, however, we measure the performance of CLQA approaches on *hard* answers.

**Inductive Query Answering.** In the transductive CLQA setup, the training and inference graphs are the same and share the same set of entities and relations, *i.e.*, $\mathcal{G}_{train} = \mathcal{G}_{inf}$ meaning that inference queries operate on the same graph, the same set of constants Con and relations. This allows query answering models to learn hardcoded entity and relation embeddings at the same time losing the capabilities to generalize to new graphs at test time.

In the inductive entity $(e)$ setup considered in Galkin et al. [14], the inference graph extends the training graph $\mathcal{G}_{train} \subset \mathcal{G}_{inf}$ but the set of relations is still fixed. Therefore, the proposed models are still bound to a certain hardcoded set of relations and cannot generalize to any arbitrary KG.

In this work, we lift all the restrictions on the training and inference graphs' vocabularies and consider the most general, inductive $(e, r)$ case when $\mathcal{G}_{inf} \neq \mathcal{G}_{train}$ and the inference graph might contain a completely different set of entities and relation types. Furthermore, missing links still have to be predicted in the inference graphs to reach *hard* answers.

**Labeling Trick GNNs.** Labeling tricks (as coined by Zhang et al. [37]) are featurization strategies in graphs for breaking symmetries in node representations which are particularly pronounced in link prediction and KG completion tasks. In the presence of such node symmetries (*automorphisms*), classical uni- and multi-relational GNN encoders [21, 33, 32] assign different *automorphic* nodes the same feature making them indistinguishable for downstream tasks. In multi-relational graphs, NBFNet [42] and A*Net [44] apply a labeling trick by using the indicator function INDICATOR$(h, v, r)$ that puts a query vector $\boldsymbol{r}$ on a head node $h$ and puts the zeros vector on other nodes $v$. The indicator function does not require entity embeddings and such models can naturally generalize to new entities (while the set of relation types is still fixed). Theoretically, such a labeling strategy learns *conditional node representations* and is provably more powerful [20] than node-level GNN encoders. In CLQA, only GNN-QE [43] applies NBFNet as a projection operator making it the only approach generalizable to the inductive $(e)$ setup [14] with new nodes at inference time. This work leverages labeling trick GNNs to generalize CLQA to arbitrary KGs with any entity and relation vocabulary.

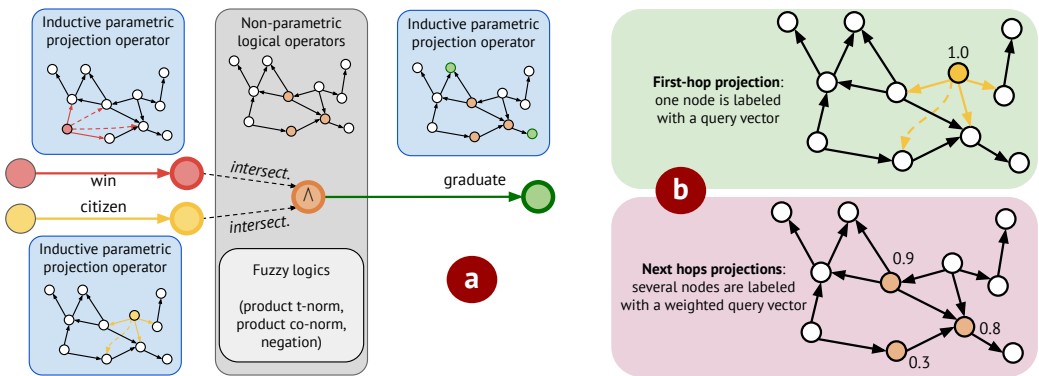

Figure 3: **(a)** Example of *ip* query answering with ULTRAQUERY: the inductive parametric projection operator (Section 4.1) executes relation projections on any graph and returns a scalar score for each entity; the scores are aggregated by non-parametric logical operators (Section 4.2) implemented with fuzzy logics. Intermediate scores are used for weighted initializion of relation projections on the next hop. **(b)** The multi-source propagation issue with a pre-trained link predictor for relation projection: pre-training on *1p* link prediction is done in the single-source labeling mode (top) where only one query node is labeled with a non-zero vector; complex queries at later intermediate hops might have several plausible sources with non-zero initial weights (bottom) where a pre-trained operator fails.

## 4 Method

We aim at designing a single foundation model for CLQA on any KG in the zero-shot fashion, *i.e.*, without training on a target graph. In the CLQA literature [19, 26, 25, 3, 43], it is common to break down query execution into a *relation projection* to traverse graph edges and predict missing links, and *logical operators* that model conjunction, disjunction, and union. The main challenge boils down to designing inductive projection and logical operators suitable for any entity and relation vocabulary.

### 4.1 Inductive Relation Projection

The vast majority of CLQA methods are inherently transductive and implement relation projections as functions over entity and relation embeddings fixed to a certain KG vocabulary, *e.g.*, with scoring functions from KG completion methods [19, 3, 5], geometric functions [26, 40], or pure neural methods [2, 34]. The only method inductive to new entities [43] learns relation embeddings and uses those as a labeling trick (Section 3) for a GNN that implements the projection operator.

As fixed relation embeddings do not transfer to new KGs with new relations, we adapt ULTRA [15], an inductive approach that builds relation representations dynamically using the invariance of *relation interactions*, as the backbone of the relation projection operator thanks to its good zero-shot performance on simple KG completion tasks across a variety of graphs. ULTRA leverages theoretical findings in multi-relational link prediction [6, 20] and learns relation representations from a *meta-graph* of relation interactions[3]. The meta-graph includes four learnable edge types or meta-relations (*head-to-tail*, *tail-to-head*, *head-to-head*, *tail-to-tail*) which are independent from KG's relation vocabulary and therefore transfer across any graph. Practically, given a graph $\mathcal{G}$ and projection query $(h, r, ?)$, ULTRA employs labeling trick GNNs on two levels. First, it builds a meta-graph $\mathcal{G}_r$ of relation interactions (a graph of relations where each node is a unique edge type in $\mathcal{G}$) and applies a labeling trick to initialize the query node $r$. Note that $|\mathcal{R}| \ll |\mathcal{E}|$, the number of unique relations is much smaller than number of entities in any KG, so processing this graph of relations introduces a rather marginal computational overhead. Running a message passing GNN over $\mathcal{G}_r$ results in *conditional relation representation* which are used as initial edge type features in the second, entity-level GNN. There, a starting node $h$ is initialized with a query vector from the obtained relation representations and running another GNN over the entity graph (with a final sigmoid readout) returns a scalar score in $[0, 1]$ representing a probability of each node to be a tail of a query $(h, r, ?)$.

The only learnable parameters in ULTRA are four meta-relations for the graph of relations and GNN weights. The four meta-relations represent structural patterns and can be mined from any

---

[3]The meta-graph can be efficiently obtained from any KG.

multi-relational KG independent of their entity and relation vocabulary. GNN weights are optimized during pre-training. Since the model does not rely on any KG-specific entity or relation vocabulary, a single pre-trained ULTRA model can be used as a zero-shot relation projection operator on any KG. Figure 3(a) illustrates the *intersection-projection* query execution process where each projection step is tackled by the same inductive projection operator with initialization depending on the start anchor node or intermediate variables.

**The multi-source propagation issue.** While it is tempting to leverage ULTRA pre-trained on multiple KG datasets for relation projection, there is a substantial distribution shift (Figure 3(b)) between KG completion and CLQA. Specifically, KG completion is a special case of relation projection where the input always contains a single node. By comparison, in multi-hop complex queries, several likely nodes might have high intermediate scores and will be labeled with non-zero vectors leading to the *multiple sources* propagation mode where a pre-trained operator is likely to fail. To alleviate the issue, we experimentally study two strategies: (1) short fine-tuning of the pre-trained projection operator on complex queries (used in the main ULTRAQUERY model), or (2) use the frozen pre-trained operator and threshold intermediate scores setting all scores below $0 < k < 1$ to zero (denoted as ULTRAQUERY LP). The insight is to limit the propagation to one or a few source nodes, thereby reducing the discrepancy between training and test distributions.

## 4.2 Inductive Logical Operations

Learnable logical operators parameterized by neural nets in many CLQA approaches [19, 26, 40, 2] fit a particular embedding space and are not transferable. Instead, we resort to differentiable but non-parametric *fuzzy logics* [31] that implement logical operators as algebraic operations (*t-norms* for conjunction and *t-conorms* for disjunction) in a bounded space $[0, 1]$ and are used in several neuro-symbolic CLQA approaches [3, 43, 4, 5, 36]. ULTRAQUERY employs fuzzy logical operators over *fuzzy sets* $\boldsymbol{x} \in [0, 1]^{|\mathcal{V}|}$ as the relation projection operator assigns a scalar in range $[0, 1]$ for each entity in a graph. The choice of a fuzzy logic is often a hyperparameter although van Krieken et al. [31] show that the *product logic* is the most stable. In product logic, given two fuzzy sets $\boldsymbol{x}, \boldsymbol{y}$, conjunction is element-wise multiplication $\boldsymbol{x} \odot \boldsymbol{y}$ and disjunction is $\boldsymbol{x} + \boldsymbol{y} - \boldsymbol{x} \odot \boldsymbol{y}$. Negation is often implemented as $\mathbf{1} - \boldsymbol{x}$ where $\mathbf{1}$ is the *universe* vector of all ones. For second- and later $i$-th hop projections, we obtain initial node states $\boldsymbol{h}_v$ by weighting a query vector $\boldsymbol{r}_i$ with their probability score $x_v$ from the fuzzy set of a previous step: $\boldsymbol{h}_v = x_v \boldsymbol{r}_i$.

## 4.3 Training

Following existing works [25, 43], ULTRAQUERY is trained on complex queries to minimize the binary cross entropy loss

$$\mathcal{L} = -\frac{1}{|\mathcal{A}_q|} \sum_{a \in \mathcal{A}_q} \log p(a|q) - \frac{1}{|\mathcal{V} \backslash \mathcal{A}_q|} \sum_{a' \in \mathcal{V} \backslash \mathcal{A}_q} \log(1 - p(a'|q)) \tag{1}$$

where $\mathcal{A}_q$ is the answer to the query $q$ and $p(a|q)$ is the probability of entity $a$ in the final output fuzzy set. ULTRAQUERY LP uses a frozen checkpoint from KG completion and is not trained on complex logical queries.

## 5 Experiments

Our experiments focus on the following research questions: (1) How does a single ULTRAQUERY model perform in the zero-shot inference mode on unseen graphs and queries compared to the baselines? (2) Does ULTRAQUERY retain the quality metrics like *faithfullness* and identify easy answers reachable by traversal? (3) How does the multi-source propagation issue affect the performance?

### 5.1 Setup and Datasets

**Datasets.** We employ 23 different CLQA datasets each with 14 standard query types and its own underlying KG with different sets of entities and relations. Following Section 3, we categorize the datasets into three groups (more statistics of the datasets and queries are provided in Appendix A):

- *Transductive* (3 datasets) where training and inference graphs are the same ($\mathcal{G}_{train} = \mathcal{G}_{inf}$) and test queries cover the same set of entities and relations: FB15k237, NELL995 and FB15k all from Ren and Leskovec [25] with at most 100 answers per query.

- *Inductive entity* ($e$) (9 datasets) from Galkin et al. [14] where inference graphs extend training graphs ($\mathcal{G}_{train} \subset \mathcal{G}_{inf}$) being up to 550% larger in the number of entities. The set of relations is fixed in each training graph and does not change at inference making the setup inductive with respect to the entities. Training queries might have more true answers in the extended inference graph.

- *Inductive entity and relation* ($e, r$) (11 datasets): we sampled a novel suite of WikiTopics-QA datasets due to the absence of standard benchmarks evaluating the hardest inductive setup where inference graphs have both new entities and relations ($\mathcal{G}_{train} \neq \mathcal{G}_{inf}$). The source graphs were adopted from the WikiTopics datasets [16], we follow the *BetaE setting* when sampling 14 query types with at most 100 answers. More details on the dataset creation procedure are in Appendix A.

**Implementation and Training.** ULTRAQUERY was trained on one FB15k237 dataset with complex queries for 10,000 steps with batch size of 32 on 4 RTX 3090 GPUs for 2 hours (8 GPU-hours in total). We initialize the model weights with an available checkpoint of ULTRA reported in Galkin et al. [15]. Following the standard setup in the literature, we train the model on 10 query types and evaluate on all 14 patterns. We employ *product t-norm* and *t-conorm* as non-parametric fuzzy logic operators to implement conjunction ($\wedge$) and disjunction ($\vee$), respectively, and use a simple $1 - x$ negation. For the ablation study, ULTRAQUERY LP uses the same frozen checkpoint (pre-trained on simple *1p* link prediction) with scores thresholding to alleviate the multi-source propagation issue (Section 4.1). More details on all hyperparameters are available in Appendix B.

**Evaluation Protocol.** As we train an ULTRAQUERY model only on one FB15k237 dataset and run zero-shot inference on other 22 graphs, the inference mode on those is *inductive* ($e, r$) since their entity and relation vocabularies are all different from the training set.

As common in the literature [25, 27], the answer set of each query is split into *easy* and *hard* answers. Easy answers are reachable by graph traversal and do not require inferring missing links whereas hard answers are those that involve at least one edge to be predicted at inference. In the rank-based evaluation, we only consider ranks of *hard* answers and filter out easy ones and report filtered Mean Reciprocal Rank (MRR) and Hits@10 as main performance metrics.

Other qualitative metrics include: (1) *faithfullness* [28], *i.e.*, the ability to recover *easy* answers reachable by graph traversal. Here, we follow the setup in Galkin et al. [14] and measure the performance of training queries on larger inference graphs where the same queries might have new true answers; (2) the ROC AUC score to estimate whether a model ranks easy answers higher than hard answers – we compute ROC AUC over *unfiltered* scores of easy answers as positive labels and hard answers as negative. (3) Mean Absolute Percentage Error (MAPE) [43] between the number of answers extracted from model's predictions and the number of ground truth answers (easy and hard combined) to estimate whether CLQA models can predict the cardinality of the answer set.

**Baselines.** In transductive and inductive ($e$) datasets, we compare a single ULTRAQUERY model with the best reported models trained end-to-end on each graph (denoted as *Best baseline* in the experiments): QTO [5] for 3 transductive datasets (FB15k237, FB15k, and NELL995) and GNN-QE [14] for 9 inductive ($e$) datasets. While a single ULTRAQUERY model has 177k parameters, the baselines are several orders of magnitude larger with a parameters count depending on the number of entities and relations, *e.g.*, a QTO model on FB15k237 has 30M parameters due to having $2000d$ entity and relation embeddings, and GNN-QE on a reference FB 175% inductive ($e$) dataset has 2M parameters. For a newly sampled suite of 11 inductive ($e, r$) datasets, we compare against the edge-type heuristic baseline introduced in Galkin et al. [14]. The heuristic selects the candidate nodes with the same incoming relation as the last hop of the query. More details on the baselines are reported in Appendix B

## 5.2 Main Experiment: Zero-shot Query Answering

In the main experiment, we measure the zero-shot query answering performance of ULTRAQUERY trained on a fraction of complex queries of one FB15k237 dataset. Figure 1 and Table 2 illustrate the comparison with the best available baselines and ablated ULTRAQUERY LP model on 23 datasets split into three categories (transductive, inductive ($e$), and inductive ($e, r$)). For each dataset, we

Table 2: Zero-shot inference results of UltraQuery and ablated UltraQuery LP on 23 datasets compared to the best reported baselines. UltraQuery was trained on one transductive FB15k237 dataset, UltraQuery LP was only pre-trained on KG completion and uses scores thresholding. The *no thrs.* version does not use any thresholding of intermediate scores (Section 4.1). The best baselines are trainable on each transductive and inductive $(e)$ dataset, and the non-parametric heuristic baseline on inductive $(e, r)$ datasets.

| Model | Inductive $(e, r)$ (11 datasets) | | | | Inductive $(e)$ (9 datasets) | | | | Transductive (3 datasets) | | | | Total Average (23 datasets) | | | |
|---|---|---|---|---|---|---|---|---|---|---|---|---|---|---|---|---|
| | EPFO avg | | neg avg | | EPFO avg | | neg avg | | EPFO avg | | neg avg | | EPFO avg | | neg avg | |
| | MRR | H@10 | MRR | H@10 | MRR | H@10 | MRR | H@10 | MRR | H@10 | MRR | H@10 | MRR | H@10 | MRR | H@10 |
| Best baseline | 0.014 | 0.029 | 0.004 | 0.007 | **0.328** | **0.469** | **0.176** | **0.297** | **0.468** | **0.603** | **0.259** | **0.409** | 0.196 | 0.276 | 0.105 | 0.173 |
| UltraQuery 0-shot | **0.280** | 0.380 | **0.193** | **0.288** | 0.312 | **0.467** | 0.139 | 0.262 | 0.411 | 0.517 | 0.240 | 0.352 | **0.309** | **0.432** | **0.178** | **0.286** |
| UltraQuery LP 0-shot | 0.268 | **0.409** | 0.104 | 0.181 | 0.277 | 0.441 | 0.098 | 0.191 | 0.322 | 0.476 | 0.150 | 0.263 | 0.279 | 0.430 | 0.107 | 0.195 |
| UltraQuery LP no thrs. | 0.227 | 0.331 | 0.080 | 0.138 | 0.246 | 0.390 | 0.085 | 0.167 | 0.281 | 0.417 | 0.127 | 0.223 | 0.242 | 0.367 | 0.088 | 0.161 |

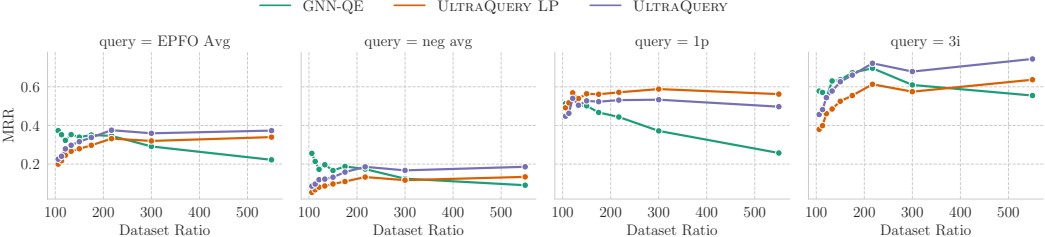

Figure 4: Mitigation of the multi-source message passing issue (Section 4) with UltraQuery: while UltraQuery LP (pre-trained only on 1p link prediction) does reach higher 1p query performance (center right), it underperforms on negation queries (center left). UltraQuery adapts to the multi-source message passing scheme and trades a fraction of 1p query performance for better averaged EPFO, *e.g.*, on the *3i* query (right), and negation queries performance. More results are in Appendix C.

measure the average MRR on 9 EPFO queries with projection, intersection, and union operators, and 5 negation queries with the negation operator, respectively.

Averaged across 23 datasets, UltraQuery outperforms available baselines by relative 50% in terms of MRR and Hits@10 on EPFO and 70% on negation queries (*e.g.*, 0.31 vs 0.20 MRR on EPFO queries and 0.178 vs 0.105 on negation queries). The largest gains are achieved on the hardest inductive $(e, r)$ datasets where the heuristic baseline is not able to cope with the task. On inductive $(e)$ datasets, UltraQuery outperforms the trainable SOTA GNN-QE model on larger inductive inference graphs and performs competitively on smaller inductive versions. On transductive benchmarks, UltraQuery lags behind the SOTA QTO model which is expected and can be attributed to the sheer model size difference (177k of UltraQuery vs 30M of QTO) and the computationally expensive brute-force approach of QTO that materializes the whole $(\mathcal{V} \times \mathcal{V} \times \mathcal{R})$ 3D tensor of scores of all possible triples. Pre-computing such tensors on three datasets takes considerable space and time, *e.g.*, 8 hours for FB15k with heavy sparsification settings to fit onto a 24 GB GPU. Still, UltraQuery outperforms a much larger QTO model on the FB15k dataset on both EPFO and negation queries. The graph behind the NELL995 dataset is a collection of disconnected components which is disadvantageous for GNNs.

We note a decent performance of UltraQuery LP trained only on simple *1p* link prediction and imbued with score thresholding to alleviate the multi-source message passing issue described in Section 4.1. Having a deeper look at other qualitative metrics in the following section, we reveal more sites where the issue incurs negative effects.

### 5.3 Analysis

Here, we study four aspects of model performance: the effect of the multi-source message passing issue mentioned in Section 4.1, the ability to recover answers achievable by edge traversal (*faithfullness*), the ability to rank easy answers higher than hard answers, and the ability to estimate the cardinality of the answer set.

**The multi-source message passing effect.** The pre-trained ULTRA checkpoint used in UltraQuery LP is tailored for singe-source message passing and struggles in the CLQA setup on later hops with several initialized nodes (Table 2). Training UltraQuery on complex queries alleviates this issue as

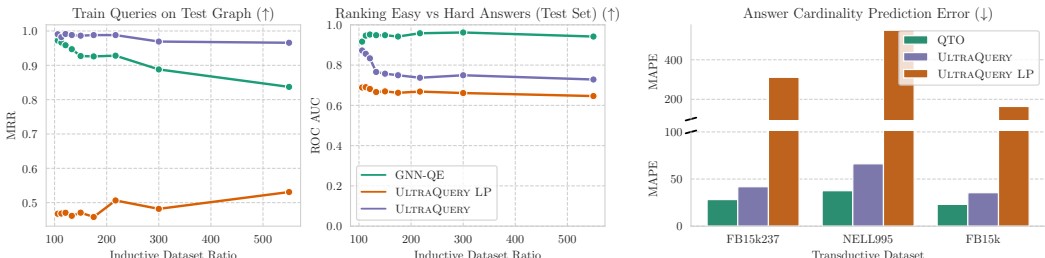

Figure 5: Qualitative analysis on 9 inductive $(e)$ and 3 transductive datasets averaged across all 14 query types. **Faithfullness, MRR (left):** ULTRAQUERY successfully finds easy answers in larger inference graphs and outperforms trained GNN-QE baselines. **Ranking of easy vs hard answers, ROC AUC (center):** zero-shot inference methods slightly lag behind trainable GNN-QE due to assigning higher scores to hard answers. **Cardinality Prediction, MAPE (right):** ULTRAQUERY is comparable to a much larger trainable baseline QTO. In all cases, ULTRAQUERY LP is significantly inferior to the main model.

shown in Figure 4, *i.e.*, while *1p* performance of ULTRAQUERY LP is higher, the overall performance on EPFO and negative queries is lacking. In contrast, ULTRAQUERY trades a fraction of *1p* single-source performance to a much better performance on negative queries (about $2\times$ improvement) and better performance on many EPFO queries, for example, on *3i* queries. Besides that, we note that the zero-shot performance of both ULTRAQUERY models does not deteriorate from the increased size of the inference graph compared to the baseline GNN-QE.

**Recovering easy answers on any graph.** *Faithfullness* [28] is the ability of a CLQA model to return *easy* query answers, *i.e.*, the answers reachable by edge traversal in the graph without predicting missing edges. While faithfullness is a common problem for many CLQA models, Figure 5 demonstrates that ULTRAQUERY almost perfectly recovers easy answers on any graph size even in the zero-shot inference regime in contrast to the best baseline. Simple score thresholding does not help ULTRAQUERY LP to deal with complex queries as all easy intermediate nodes have high scores above the threshold and the multi-source is more pronounced.

**Ranking easy and hard answers.** A reasonable CLQA model is likely to score easy answers higher than hard ones that require inferring missing links [14]. Measuring that with ROC AUC (Figure 5), ULTRAQUERY is behind the baseline due to less pronounced decision boundaries (overlapping distributions of scores) between easy and hard answers' scores. Still, due to scores filtering when computing ranking metrics, this fact does not have a direct negative impact on the overall performance.

**Estimating the answer set cardinality.** Neural-symbolic models like GNN-QE and QTO have the advantage of estimating the cardinality of the answer set based on the final scores without additional supervision. As shown in Figure 5, ULTRAQUERY is comparable to the larger and trainable QTO baseline on FB15k237 (on which the model was trained) as well as on other datasets in the zero-shot inference regime. Since cardinality estimation is based on score thresholding, ULTRAQUERY LP is susceptible to the multi-source propagation issue with many nodes having a high score and is not able to deliver a comparable performance.

**Varying the number of graphs in training.** Figure 6 and Table 3 report the inductive inference CLQA performance depending on the number of KGs in the training mixture. The original ULTRAQUERY was trained on queries from the FB15k237. In order to maintain the zero-shot inductive inference setup on 11 inductive $(e, r)$ and 9 inductive $(e)$ datasets, we trained new model versions on the rest of BetaE datasets, that is, ULTRAQUERY 2G combines FB15k237 and NELL995 queries (trained for 20k steps), ULTRAQUERY 3G combines FB15k273, NELL995, and FB15k queries (trained for 30k steps). The most noticeable improvement of 2G and 3G versions is the increased MRR and Hits@10 on EPFO queries (9 query

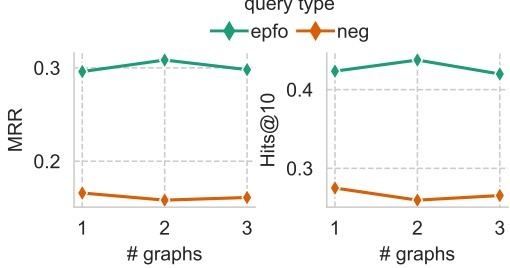

Figure 6: Average MRR (left) and Hits@10 (right) of 9 inductive $(e)$ and 11 inductive $(e, r)$ CLQA datasets for EPFO and negation queries depending on the number of graphs in the training mix.

Table 3: Zero-shot inference results (on 20 inductive datasets) of ULTRAQUERY trained on 1, 2, and 3 datasets, respectively. The biggest gains of the 2G model are **in bold**.

| Model | Inductive $(e, r)$ (11 datasets) | | | | Inductive $(e)$ (9 datasets) | | | | Total Average (20 datasets) | | | |
|---|---|---|---|---|---|---|---|---|---|---|---|---|
| | EPFO avg | | neg avg | | EPFO avg | | neg avg | | EPFO avg | | neg avg | |
| | MRR | H@10 | MRR | H@10 | MRR | H@10 | MRR | H@10 | MRR | H@10 | MRR | H@10 |
| ULTRAQUERY 1G | 0.280 | 0.380 | 0.193 | 0.288 | 0.312 | 0.467 | 0.139 | 0.262 | 0.296 | 0.423 | 0.166 | 0.275 |
| ULTRAQUERY 2G | **0.310** | **0.413** | 0.187 | 0.275 | 0.307 | 0.463 | 0.130 | 0.244 | 0.308 | 0.438 | 0.158 | 0.260 |
| ULTRAQUERY 3G | 0.304 | 0.402 | 0.195 | 0.292 | 0.292 | 0.438 | 0.127 | 0.239 | 0.298 | 0.420 | 0.161 | 0.265 |

types) on 11 inductive $(e, r)$ datasets yielding about 10% gains. On 9 inductive $(e)$ datasets the performance is either on par with the 1G version or a bit lower. Averaged across 20 datasets, the 2G version exhibits the best EPFO performance at the cost of slightly reduced negation query performance.

## 6 Discussion and Future Work

**Limitations.** The only parameterized component of ULTRAQUERY is the projection operator and, therefore, limitations and improvement opportunities stem from the projection operator [15] and its interplay with the multi-hop query answering framework. For instance, new mechanisms of tackling the multi-source propagation, better pre-training strategies, and scaling might positively impact the zero-shot CLQA performance. The support for very large KGs could be further improved by adopting more scalable entity-level GNN predictors like A*Net [44] or AdaProp [39] which have been shown to scale to graphs of millions of nodes. We are optimistic that ULTRAQUERY could scale to such graphs when integrated with those models.

**Conclusion and Future Work.** We presented ULTRAQUERY, the first foundation model for inductive zero-shot complex logical query answering on any KG that combines a parameterized, inductive projection operator with non-parametric logical operators. Alleviating the multi-source message propagation issue is the key to adapt pre-trained projection operators into the multi-hop query answering framework. ULTRAQUERY performs comparably to or better than strong baselines trained specifically on each graph and at the same time retains key qualitative features like faithfullness and answer cardinality estimation. Having a single query answering model working on any KG, the scope for future work is vast as highlighted by Ren et al. [27] and includes, for example, better theoretical understanding of logical expressiveness bounds, supporting more query patterns beyond simple trees [35, 36], queries without anchor nodes [7], hyper-relational queries [1], queries with numerical literals [11], or temporal queries [23].

**Impact Statement.** We do not envision direct ethical or societal consequences of this work. Still, models capable of zero-shot inference on any graph might be applied to domains other than those designed by the authors. Positive impacts include saving compute resources and reducing carbon footprint of training specific models tailored for each graph.

## Acknowledgements

This project is supported by Intel-Mila partnership program, the Natural Sciences and Engineering Research Council (NSERC) Discovery Grant, the Canada CIFAR AI Chair Program, collaboration grants between Microsoft Research and Mila, Samsung Electronics Co., Ltd., Amazon Faculty Research Award, Tencent AI Lab Rhino-Bird Gift Fund and a NRC Collaborative R&D Project (AI4D-CORE-06). This project was also partially funded by IVADO Fundamental Research Project grant PRF-2019-3583139727. The computation resource of this project is supported by Mila[4], Calcul Québec[5] and the Digital Research Alliance of Canada[6].

This work was funded in part by the National Science Foundation (NSF) awards, CCF-1918483, CAREER IIS-1943364 and CNS-2212160, Amazon Research Award, AnalytiXIN, and the Wabash Heartland Innovation Network (WHIN). Computing infrastructure was supported in part by CNS-1925001 (CloudBank). Any opinions, findings and conclusions or recommendations expressed in this material are those of the authors and do not necessarily reflect the views of the sponsors.

---

[4] https://mila.quebec/
[5] https://www.calculquebec.ca/
[6] https://alliancecan.ca/

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

## A Datasets

### A.1 Dataset statistics

First, Table 4, Table 5, and Table 6 provide the necessary details on the graphs behind the CLQA datasets. Then, Table 7, Table 8, and Table 9 list the query statistics. Transductive datasets are BetaE [25] datasets (MIT license), inductive $(e)$ datasets are adopted from Galkin et al. [14] (CC BY 4.0 license) where validation and test inference graphs extend the training graph. The ratio denotes the size of the inference graph to the size of the training graph (in the number of nodes), that is, $\mathcal{V}_{inf}/\mathcal{V}_{train}$. In the following Appendix A.2 we provide more details in sampling 11 new inductive $(e, r)$ datasets WikiTopics-CLQA (available under the CC BY 4.0 license).

Table 4: Graph in transductive datasets (3) from Ren and Leskovec [25]. Inverse triples and edge types are included in the splits. Train, Valid, Test denote triples in the respective set.

| Dataset | Entities | Rels | Train | Valid | Test |
|---|---|---|---|---|---|
| FB15k | 14,951 | 2,690 | 966,284 | 100,000 | 118,142 |
| FB15k237 | 14,505 | 474 | 544,230 | 35,052 | 40,876 |
| NELL995 | 63,361 | 400 | 228,426 | 28,648 | 28,534 |

Table 5: Graphs in inductive $(e)$ datasets (9) from Galkin et al. [14]. Inverse triples and edge types are included in all graphs. Validation and Test splits contain an inference graph $(\mathcal{V}_{inf}, \mathcal{E}_{inf})$ which is a superset of the training graph with new nodes, and missing edges to predict (Valid and Test, respectively).

| Ratio, % | Rels | Training Graph | | Validation Graph | | | Test Graph | | |
|---|---|---|---|---|---|---|---|---|---|
| | | Entities | Triples | Entities | Triples | Valid | Entities | Triples | Test |
| 106% | 466 | 13,091 | 493,425 | 13,801 | 551,336 | 10,219 | 13,802 | 538,896 | 8,023 |
| 113% | 468 | 11,601 | 401,677 | 13,022 | 491,518 | 15,849 | 13,021 | 486,068 | 14,893 |
| 122% | 466 | 10,184 | 298,879 | 12,314 | 413,554 | 20,231 | 12,314 | 430,892 | 23,289 |
| 134% | 466 | 8,634 | 228,729 | 11,468 | 373,262 | 25,477 | 11,471 | 367,810 | 24,529 |
| 150% | 462 | 7,232 | 162,683 | 10,783 | 311,462 | 26,235 | 10,782 | 331,352 | 29,755 |
| 175% | 436 | 5,560 | 102,521 | 9,801 | 265,412 | 28,691 | 9,781 | 266,494 | 28,891 |
| 217% | 446 | 4,134 | 52,455 | 9,062 | 227,284 | 30,809 | 9,058 | 212,386 | 28,177 |
| 300% | 412 | 2,650 | 24,439 | 8,252 | 178,680 | 27,135 | 8,266 | 187,156 | 28,657 |
| 550% | 312 | 1,084 | 5,265 | 7,247 | 136,558 | 22,981 | 7,275 | 133,524 | 22,503 |

Table 6: Graphs in the newly sampled inductive entity and relation $(e, r)$ WikiTopics-CLQA datasets (11). Triples denote the number of edges of the graph given at training, validation, or test. Valid and Test denote triples to be predicted in the validation and test sets in the respective validation and test graph.

| Dataset | Training Graph | | | Validation Graph | | | | Test Graph | | | |
|---|---|---|---|---|---|---|---|---|---|---|---|
| | Entities | Rels | Triples | Entities | Rels | Triples | Valid | Entities | Rels | Triples | Test |
| Art | 10000 | 65 | 27262 | 10000 | 65 | 27262 | 3026 | 10000 | 65 | 28023 | 3113 |
| Award | 10000 | 17 | 23821 | 10000 | 13 | 23821 | 2646 | 10000 | 17 | 25056 | 2783 |
| Education | 10000 | 19 | 14355 | 10000 | 19 | 14355 | 1594 | 10000 | 19 | 14193 | 1575 |
| Health | 10000 | 31 | 15539 | 10000 | 31 | 15539 | 1725 | 10000 | 31 | 15337 | 1703 |
| Infrastructure | 10000 | 37 | 21990 | 10000 | 37 | 21990 | 2443 | 10000 | 37 | 21646 | 2405 |
| Location | 10000 | 62 | 85063 | 10000 | 62 | 85063 | 9451 | 10000 | 62 | 80269 | 8917 |
| Organization | 10000 | 34 | 33325 | 10000 | 34 | 33325 | 3702 | 10000 | 34 | 31314 | 3357 |
| People | 10000 | 40 | 55698 | 10000 | 40 | 55698 | 6188 | 10000 | 40 | 58530 | 6503 |
| Science | 10000 | 66 | 12576 | 10000 | 66 | 12576 | 1397 | 10000 | 66 | 12516 | 1388 |
| Sport | 10000 | 34 | 47251 | 10000 | 34 | 47251 | 5250 | 10000 | 34 | 46717 | 5190 |
| Taxonomy | 10000 | 59 | 18921 | 10000 | 59 | 18921 | 2102 | 10000 | 59 | 19416 | 2157 |

### A.2 WikiTopics-CLQA

The WikiTopics dataset introduced by Gao et al. [16] was used to evaluate link prediction model's zero-shot performance in the inductive $(e, r)$ setting, i.e., when the test-time inference graph contains *both* new entities and new relations unseen in training. It grouped relations into 11 different topics, or

Table 7: Statistics of 3 transductive datasets

| Split | Query Type | FB15k | FB15k-237 | NELL995 |
|---|---|---|---|---|
| Train | 1p/2p/3p/2i/3i | 273,710 | 149,689 | 107,982 |
| | 2in/3in/inp/pin/pni | 27,371 | 14,968 | 10,798 |
| Valid | 1p | 59,078 | 20,094 | 16,910 |
| | Others | 8,000 | 5,000 | 4,000 |
| Test | 1p | 66,990 | 22,804 | 17,021 |
| | Others | 8,000 | 5,000 | 4,000 |

Table 8: Statistics of 9 inductive ($e$) datasets.

| Ratio | Graph | 1p | 2p | 3p | 2i | 3i | pi | ip | 2u | up | 2in | 3in | inp | pin | pni |
|---|---|---|---|---|---|---|---|---|---|---|---|---|---|---|---|
| 106% | training | 135,613 | 50,000 | 50,000 | 50,000 | 50,000 | 50,000 | 50,000 | 50,000 | 50,000 | 40,000 | 50,000 | 50,000 | 50,000 | 50,000 |
| | validation | 6,582 | 10,000 | 10,000 | 10,000 | 10,000 | 10,000 | 10,000 | 10,000 | 10,000 | 1,000 | 1,000 | 1,000 | 1,000 | 1,000 |
| | test | 5,446 | 10,000 | 10,000 | 10,000 | 10,000 | 10,000 | 10,000 | 10,000 | 10,000 | 1,000 | 1,000 | 1,000 | 1,000 | 1,000 |
| 113% | training | 115,523 | 50,000 | 50,000 | 50,000 | 50,000 | 50,000 | 50,000 | 50,000 | 50,000 | 40,000 | 50,000 | 50,000 | 50,000 | 50,000 |
| | validation | 10,256 | 10,000 | 10,000 | 10,000 | 10,000 | 10,000 | 10,000 | 10,000 | 10,000 | 1,000 | 1,000 | 1,000 | 1,000 | 1,000 |
| | test | 9,782 | 10,000 | 10,000 | 10,000 | 10,000 | 10,000 | 10,000 | 10,000 | 10,000 | 1,000 | 1,000 | 1,000 | 1,000 | 1,000 |
| 121% | training | 91,228 | 50,000 | 50,000 | 50,000 | 50,000 | 50,000 | 50,000 | 50,000 | 50,000 | 50,000 | 40,000 | 50,000 | 50,000 | 50,000 |
| | validation | 12,696 | 10,000 | 10,000 | 10,000 | 10,000 | 10,000 | 10,000 | 10,000 | 10,000 | 5,000 | 5,000 | 5,000 | 5,000 | 5,000 |
| | test | 14,458 | 10,000 | 10,000 | 10,000 | 10,000 | 10,000 | 10,000 | 10,000 | 10,000 | 5,000 | 5,000 | 5,000 | 5,000 | 5,000 |
| 133% | training | 75,326 | 50,000 | 50,000 | 50,000 | 50,000 | 50,000 | 50,000 | 50,000 | 50,000 | 40,000 | 50,000 | 50,000 | 50,000 | 50,000 |
| | validation | 15,541 | 50,000 | 50,000 | 50,000 | 50,000 | 50,000 | 50,000 | 20,000 | 20,000 | 20,000 | 5,000 | 5,000 | 5,000 | 5,000 |
| | test | 15,270 | 50,000 | 50,000 | 50,000 | 50,000 | 50,000 | 50,000 | 20,000 | 20,000 | 20,000 | 5,000 | 5,000 | 5,000 | 5,000 |
| 150% | training | 56,114 | 50,000 | 50,000 | 50,000 | 50,000 | 50,000 | 50,000 | 50,000 | 50,000 | 40,000 | 50,000 | 50,000 | 50,000 | 50,000 |
| | validation | 16,229 | 50,000 | 50,000 | 50,000 | 50,000 | 50,000 | 50,000 | 50,000 | 50,000 | 5,000 | 5,000 | 5,000 | 5,000 | 5,000 |
| | test | 17,683 | 50,000 | 50,000 | 50,000 | 50,000 | 50,000 | 50,000 | 50,000 | 50,000 | 5,000 | 5,000 | 5,000 | 5,000 | 5,000 |
| 175% | training | 38,851 | 50,000 | 50,000 | 50,000 | 50,000 | 50,000 | 50,000 | 50,000 | 50,000 | 40,000 | 50,000 | 50,000 | 50,000 | 50,000 |
| | validation | 17,235 | 50,000 | 50,000 | 50,000 | 50,000 | 50,000 | 50,000 | 50,000 | 50,000 | 10,000 | 10,000 | 10,000 | 10,000 | 10,000 |
| | test | 17,476 | 50,000 | 50,000 | 50,000 | 50,000 | 50,000 | 50,000 | 50,000 | 50,000 | 10,000 | 10,000 | 10,000 | 10,000 | 10,000 |
| 217% | training | 22,422 | 30,000 | 30,000 | 50,000 | 50,000 | 50,000 | 50,000 | 50,000 | 50,000 | 30,000 | 30,000 | 50,000 | 50,000 | 50,000 |
| | validation | 18,168 | 50,000 | 50,000 | 50,000 | 50,000 | 50,000 | 50,000 | 50,000 | 50,000 | 10,000 | 10,000 | 10,000 | 10,000 | 10,000 |
| | test | 16,902 | 50,000 | 50,000 | 50,000 | 50,000 | 50,000 | 50,000 | 50,000 | 50,000 | 10,000 | 10,000 | 10,000 | 10,000 | 10,000 |
| 300% | training | 11,699 | 15,000 | 15,000 | 40,000 | 40,000 | 50,000 | 50,000 | 50,000 | 50,000 | 15,000 | 15,000 | 50,000 | 40,000 | 50,000 |
| | validation | 16,189 | 50,000 | 50,000 | 50,000 | 50,000 | 50,000 | 50,000 | 50,000 | 50,000 | 10,000 | 10,000 | 10,000 | 10,000 | 10,000 |
| | test | 17,105 | 50,000 | 50,000 | 50,000 | 50,000 | 50,000 | 50,000 | 50,000 | 50,000 | 10,000 | 10,000 | 10,000 | 10,000 | 10,000 |
| 550% | training | 3,284 | 15,000 | 15,000 | 40,000 | 40,000 | 50,000 | 50,000 | 50,000 | 50,000 | 10,000 | 10,000 | 30,000 | 30,000 | 30,000 |
| | validation | 13,616 | 50,000 | 50,000 | 50,000 | 50,000 | 50,000 | 50,000 | 50,000 | 50,000 | 10,000 | 10,000 | 10,000 | 10,000 | 10,000 |
| | test | 13,670 | 50,000 | 50,000 | 50,000 | 50,000 | 50,000 | 50,000 | 50,000 | 50,000 | 10,000 | 10,000 | 10,000 | 10,000 | 10,000 |

Table 9: WikiTopics-CLQA statistics: the number of queries generated per query pattern for each topic knowledge graph of WikiTopics [16]. Numbers are the same for both the training and inference graph. We follow the same 14 query patterns introduced by Ren and Leskovec [25].

| Topics | 1p | 2p | 3p | 2i | 3i | pi | ip | 2in | 3in | pin | pni | inp | 2u | up |
|---|---|---|---|---|---|---|---|---|---|---|---|---|---|---|
| Art | 3113 | 10000 | 10000 | 10000 | 10000 | 10000 | 10000 | 1000 | 1000 | 1000 | 1000 | 1000 | 10000 | 10000 |
| Award | 2783 | 10000 | 10000 | 10000 | 10000 | 10000 | 10000 | 1000 | 1000 | 1000 | 1000 | 1000 | 10000 | 10000 |
| Education | 1575 | 10000 | 10000 | 10000 | 10000 | 10000 | 10000 | 1000 | 1000 | 1000 | 1000 | 1000 | 10000 | 10000 |
| Health | 1703 | 10000 | 10000 | 10000 | 10000 | 10000 | 10000 | 1000 | 1000 | 1000 | 1000 | 1000 | 10000 | 10000 |
| Infrastructure | 2405 | 10000 | 10000 | 10000 | 10000 | 10000 | 10000 | 1000 | 1000 | 1000 | 1000 | 1000 | 10000 | 10000 |
| Location | 8000 | 8917 | 4000 | 8000 | 8000 | 8000 | 8000 | 1000 | 1000 | 1000 | 1000 | 1000 | 8000 | 8000 |
| Organization | 3357 | 8000 | 4000 | 8000 | 8000 | 8000 | 8000 | 1000 | 1000 | 1000 | 1000 | 1000 | 8000 | 8000 |
| People | 6503 | 10000 | 10000 | 10000 | 10000 | 10000 | 10000 | 1000 | 1000 | 1000 | 1000 | 1000 | 10000 | 10000 |
| Science | 1388 | 10000 | 10000 | 10000 | 10000 | 10000 | 10000 | 1000 | 1000 | 1000 | 1000 | 1000 | 10000 | 10000 |
| Sport | 5190 | 8000 | 4000 | 8000 | 8000 | 8000 | 8000 | 1000 | 1000 | 1000 | 1000 | 1000 | 8000 | 8000 |
| Taxonomy | 2157 | 8000 | 8000 | 8000 | 8000 | 8000 | 8000 | 1000 | 1000 | 1000 | 1000 | 1000 | 8000 | 8000 |

domains, such as art, education, health care, and sport. Two graphs, $\mathcal{G}_{train}^{(T)}$ and $\mathcal{G}_{inf}^{(T)}$ along with the missing triples $\mathcal{E}_{valid}^{(T)}$ and $\mathcal{E}_{test}^{(T)}$, were provided for each topic $T$, which had the same set of relations but different (potentially overlapping) set of entities. The goal was to train models on the training graphs $\mathcal{G}_{train}^{(T)}$ of some topic $T$, and test on the inference graph $\mathcal{G}_{inf}^{(T')}$ of an unseen topic $T'$. The model's validation performance was evaluated on the missing triples $\mathcal{E}_{valid}^{(T)}$ when observing training graph $\mathcal{G}_{train}^{(T)}$ as inputs, and its test performance was evaluated on $\mathcal{E}_{test}^{(T')}$ when observing the test inference graph $\mathcal{G}_{inf}^{(T')}$ as inputs. Table 6 shows the statistics of the 11 topic-specific knowledge graphs in WikiTopics.

We follow the procedures in BetaE [25] to generate queries and answers of the 14 query patterns using the knowledge graphs in WikiTopics. We name the resulting dataset WikiTopics-CLQA. For each topic $T$, we generate three sets of queries and answers (training, validation, test), using the training graph $\mathcal{G}_{train}^{(T)}$ for training and validation, and inference graph $\mathcal{G}_{inf}^{(T)}$ for test queries, respectively.

Training queries on $\mathcal{G}_{train}^{(T)}$ only have easy answers, validation (test) set easy answers are attained by traversing $\mathcal{G}_{train}^{(T)}$ ($\mathcal{G}_{inf}^{(T)}$), whereas the full set of answers (easy and hard) are attained by traversing the graph $\mathcal{G}_{train}^{(T)}$ merged with $\mathcal{E}_{valid}^{(T)}$ ($\mathcal{G}_{inf}^{(T)}$ merged with $\mathcal{E}_{test}^{(T)}$). Hence, the hard answers cannot be found unless the model is capable of imputing missing links. In our experiments, we only use the inference graph $\mathcal{G}_{inf}^{(T)}$ and the test queries and answers for evaluating zero-shot inference performance. Table 9 shows the statistics of the WikiTopics-CLQA dataset.

## B  Hyperparameters and Baselines

Both ULTRAQUERY and ULTRAQUERY LP are implemented with PyTorch [24] (BSD-style license) and PyTorch-Geometric [12] (MIT license). Both ULTRAQUERY and ULTRAQUERY LP are initialized from the pre-trained ULTRA checkpoint published by the authors and have the same GNN architectures with parameter count (177k). ULTRAQUERY is further trained for 10,000 steps on complex queries from the FB15k237 dataset. Hyperparameters of ULTRAQUERY and its training details are listed in Table 10.

ULTRAQUERY LP employs thresholding of intermediate fuzzy set scores as one of the ways to alleviate the multi-source propagation issue (Section 4.1). Generally, the threshold is set to 0.8 with a few exceptions:

• 0.97 in NELL995

Below, we discuss the best available baselines for each dataset family.

**Transductive** (3 datasets): QTO [5]. QTO requires $2000d$ ComplEx [30] entity and relation embeddings pre-computed for each graph, *e.g.*, taking 30M parameters on the FB15k237 graph with 15k nodes. Further, QTO materializes the whole $(\mathcal{V} \times \mathcal{V} \times \mathcal{R})$ 3D tensor of scores of all possible triples for each graph. Pre-computing such tensors on three datasets takes considerable space and time, *e.g.*, 8 hours for FB15k with heavy sparsification settings to fit onto a 24 GB GPU.

**Inductive** $(e)$ (9 datasets): GNN-QE [43]. The framework of GNN-QE is similar to ULTRAQUERY but the backbone relation projection operator is implemented with NBFNet [42] which is only inductive to entities and still learns graph-specific relation embeddings. Parameter count, therefore, depends on the number of unique relation types, *e.g.*, 2M for the FB 175% split with 436 unique relations.

**Inductive** $(e, r)$ (11 datasets): for newly sampled datasets, due to the absence of fully inductive trainable baselines, we compare against a rule-based heuristic baseline similar to the baseline in Galkin et al. [14]. The baseline looks up candidate entities that have the same incoming relation type as the current relation projection (note that the identity of the starting head node in this case is ignored). The heuristic filters entities into two classes (satisfying the incoming relation type or not), hence, in order to create a ranking, we randomly shuffle entities in each class. This baseline is non-parametric, does not require any training, and represents a sanity check of CLQA models. Still, as shown in Galkin et al. [14], the baseline might outperform certain inductive reasoning approaches parameterized by neural networks.

## C  More Results

Table 11 corresponds to Figure 1 and Table 2 and provides MRR and Hits@10 results for ULTRA-QUERY, ULTRAQUERY LP, and Best Baseline for each dataset averaged across 9 EPFO query types and 5 negation query types. Figure 7 is the full version of Figure 4 and illustrates the performance of all 3 compared models on 9 inductive $(e)$ datasets on 14 query types together with their averaged values (EPFO avg and neg avg, respectively).

Table 10: ULTRAQUERY hyperparameters. $GNN_r$ denotes a GNN over the graph of relations $\mathcal{G}_r$, $GNN_e$ is a GNN over the original entity graph $\mathcal{G}$.

| Hyperparameter | | ULTRAQUERY training |
|---|---|---|
| $GNN_r$ | # layers | 6 |
| | hidden dim | 64 |
| | message | DistMult |
| | aggeregation | sum |
| $GNN_e$ | # layers | 6 |
| | hidden dim | 64 |
| | message | DistMult |
| | aggregation | sum |
| | $g(\cdot)$ | 2-layer MLP |
| Learning | optimizer | AdamW |
| | learning rate | 0.0005 |
| | training steps | 10,000 |
| | adv temperature | 0.2 |
| | traversal dropout | 0.25 |
| | batch size | 64 |
| | training queries | FB15k237 |

Table 11: Full results (MRR, Hits@10) of ULTRAQUERY LP and ULTRAQUERY in the zero-shot inference regime on transductive, entity-inductive $(e)$, and fully inductive $(e, r)$ datasets compared to the best-reported baselines averaged across 9 EPFO query types (EPFO avg) and 5 negation query types (Negation avg). ULTRAQUERY was fine-tuned only on FB15k237 queries. The numbers correspond to Table 2 and Figure 1.

| | ULTRAQUERY LP | | | | ULTRAQUERY | | | | Best Baseline | | | |
|---|---|---|---|---|---|---|---|---|---|---|---|---|
| | EPFO avg | | Negation avg | | EPFO avg | | Negation avg | | EPFO avg | | Negation avg | |
| | MRR | Hits@10 | MRR | Hits@10 | MRR | Hits@10 | MRR | Hits@10 | MRR | Hits@10 | MRR | Hits@10 |
| | | | | | transductive datasets | | | | | | | |
| FB15k237 | 0.216 | 0.362 | 0.082 | 0.164 | 0.242 | 0.378 | 0.08 | 0.174 | 0.335 | 0.491 | 0.155 | 0.295 |
| FB15k | 0.501 | 0.672 | 0.291 | 0.465 | 0.764 | 0.834 | 0.567 | 0.725 | 0.740 | 0.837 | 0.492 | 0.664 |
| NELL995 | 0.249 | 0.395 | 0.079 | 0.160 | 0.226 | 0.341 | 0.073 | 0.159 | 0.329 | 0.483 | 0.129 | 0.268 |
| | | | | | inductive $(e)$ datasets | | | | | | | |
| FB 550% | 0.340 | 0.518 | 0.134 | 0.251 | 0.373 | 0.535 | 0.186 | 0.332 | 0.222 | 0.331 | 0.091 | 0.158 |
| FB 300% | 0.320 | 0.496 | 0.117 | 0.227 | 0.359 | 0.526 | 0.168 | 0.312 | 0.291 | 0.426 | 0.125 | 0.224 |
| FB 217% | 0.332 | 0.509 | 0.133 | 0.252 | 0.375 | 0.537 | 0.186 | 0.337 | 0.346 | 0.492 | 0.174 | 0.301 |
| FB 175% | 0.297 | 0.469 | 0.110 | 0.214 | 0.338 | 0.499 | 0.159 | 0.297 | 0.351 | 0.507 | 0.188 | 0.336 |
| FB 150% | 0.279 | 0.445 | 0.097 | 0.190 | 0.316 | 0.473 | 0.132 | 0.255 | 0.339 | 0.493 | 0.167 | 0.303 |
| FB 133% | 0.266 | 0.426 | 0.087 | 0.167 | 0.298 | 0.451 | 0.122 | 0.238 | 0.353 | 0.514 | 0.197 | 0.341 |
| FB 121% | 0.246 | 0.400 | 0.081 | 0.164 | 0.279 | 0.430 | 0.119 | 0.232 | 0.323 | 0.462 | 0.173 | 0.291 |
| FB 113% | 0.217 | 0.362 | 0.067 | 0.136 | 0.240 | 0.387 | 0.097 | 0.192 | 0.352 | 0.494 | 0.214 | 0.339 |
| FB 106% | 0.200 | 0.340 | 0.054 | 0.114 | 0.226 | 0.370 | 0.086 | 0.162 | 0.373 | 0.504 | 0.256 | 0.377 |
| | | | | | inductive $(e, r)$ datasets | | | | | | | |
| Art | 0.249 | 0.389 | 0.086 | 0.157 | 0.248 | 0.349 | 0.083 | 0.137 | 0.016 | 0.031 | 0.006 | 0.014 |
| Award | 0.224 | 0.413 | 0.046 | 0.098 | 0.227 | 0.354 | 0.152 | 0.274 | 0.004 | 0.006 | 0.002 | 0.002 |
| Edu | 0.142 | 0.258 | 0.066 | 0.122 | 0.179 | 0.249 | 0.119 | 0.176 | 0.008 | 0.014 | 0.003 | 0.005 |
| Health | 0.317 | 0.466 | 0.159 | 0.231 | 0.317 | 0.394 | 0.419 | 0.525 | 0.019 | 0.040 | 0.010 | 0.020 |
| Infrastructure | 0.392 | 0.551 | 0.087 | 0.170 | 0.337 | 0.461 | 0.235 | 0.356 | 0.010 | 0.018 | 0.003 | 0.005 |
| Location | 0.508 | 0.678 | 0.198 | 0.371 | 0.569 | 0.679 | 0.402 | 0.585 | 0.007 | 0.017 | 0.001 | 0.002 |
| Organization | 0.098 | 0.190 | 0.023 | 0.048 | 0.169 | 0.270 | 0.082 | 0.171 | 0.005 | 0.008 | 0.001 | 0.002 |
| People | 0.373 | 0.530 | 0.182 | 0.281 | 0.332 | 0.443 | 0.194 | 0.285 | 0.005 | 0.010 | 0.002 | 0.003 |
| Science | 0.206 | 0.348 | 0.048 | 0.093 | 0.158 | 0.255 | 0.071 | 0.131 | 0.041 | 0.085 | 0.010 | 0.020 |
| Sport | 0.215 | 0.357 | 0.095 | 0.166 | 0.252 | 0.371 | 0.178 | 0.269 | 0.015 | 0.030 | 0.005 | 0.008 |
| Taxonomy | 0.230 | 0.315 | 0.151 | 0.255 | 0.290 | 0.360 | 0.183 | 0.261 | 0.034 | 0.066 | 0.009 | 0.017 |

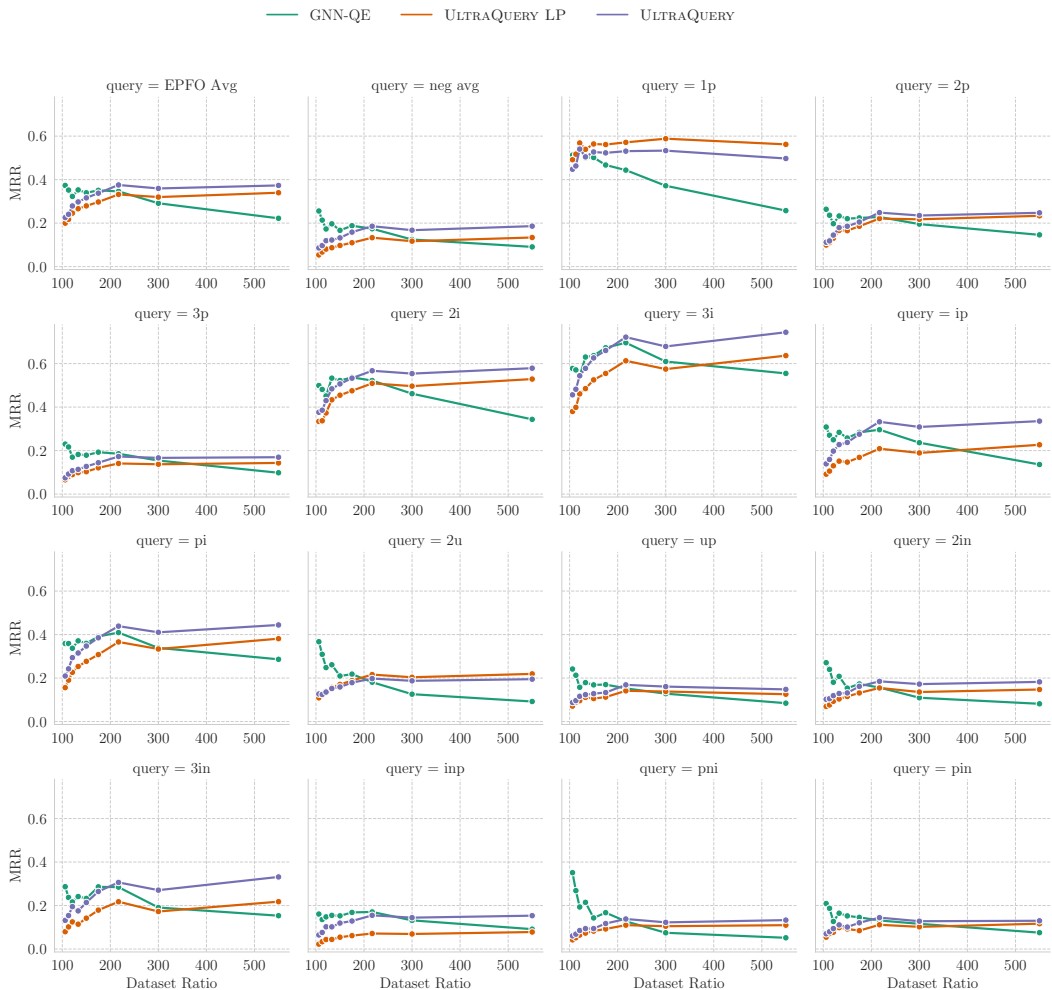

Figure 7: Full results on 9 inductive (e) datasets corresponding to Figure 4: albeit ULTRAQUERY LP outperforms the main ULTRAQUERY on simple *1p* queries, it suffers from the multi-source propagation issue on complex queries. ULTRAQUERY trades a fraction of 1p query performance for significantly better average performance on 9 EPFO and 5 negation query types with particularly noticeable gains on intersection and *2in*, *3in* queries.

