# OpenReview forum: "A Foundation Model for Zero-shot Logical Query Reasoning"
_NeurIPS.cc/2024/Conference — NeurIPS 2024 poster_

### Official Review · Reviewer_TgMu · 2024-07-05

**Soundness:** 3
**Presentation:** 3
**Contribution:** 4
**Rating:** 7
**Confidence:** 4

**Summary:**

This paper considers the inductive setting of complex logical query over incomplete knowledge graph, in which unseen entities and relations exist. To address this challenge but the particularly important setting, this paper generalizes the foundation model of knowledge graph completion Ultra and proposes UltraQuery. Experiments show that UltraQuery trained on FB15k-237 can achieve competitive performance across 22 out-of-distribution KG datasets and can zero-shot answer the logical query on other KG.

**Strengths:**

1.This paper introduces a foundation model which can answer the logical queries over unseen KG. The foundation model is trained on the FB15k-237 and achieve the competitive performance comparing with SoTA baseline. This results is quite impressive.

2. This paper is well organized and well written.

3. This discussion  is solid and the experiments well supports the foundation model for logical query answering.

**Weaknesses:**

1. I am concerning about the novelty of the proposed methods. While the results of UltraQuery is impressive, the zero-shot ability is naturally derived from the foundation model of KG completeness, Ultra. This extension reminds the extension NBF-NET to GNN-QE.

**Questions:**

1. The feature used in UltraQuery is the intersection between entity and relation, which deprecates the representation of the specified entity (relations) and can generalize the unseen entity (relation). I am wondering if UltraQuery can exploit the feature from entity (relation) like attributes, types, and so on to improve the performance by fine-tuning.

2. Will you present  the results of the setting providing in your further work? In terms of the query patterns, the results can be directedly evaluated without modifying the models.


[1] Hang Yin, Zihao Wang, Weizhi Fei, and Yangqiu Song. EFOk-cqa: Towards knowledge graph complex query answering beyond set operation. arXiv preprint arXiv:2307.13701, 2023.

[2] Hang Yin, Zihao Wang, and Yangqiu Song. Rethinking complex queries on knowledge graphs with neural link predictors. In The Twelfth International Conference on Learning Representa- tions, 2024. URL https://openreview.net/forum?id=1BmveEMNbG.

[3] Pablo Barcelo ́, Tamara Cucumides, Floris Geerts, Juan Reutter, and Miguel Romero. A neuro- symbolic framework for answering conjunctive queries. arXiv preprint arXiv:2310.04598, 2023.

---

> ### Author Rebuttal · Authors · 2024-08-06
>
> We thank the reviewer for appreciating our work highlighting the results, paper organization, and discussions. Please find our comments below:
>
> > **W1.** novelty of the proposed methods.
>
> In this work, we introduced a new setup – a fully-inductive, zero-shot generalization of complex query answering on new, unseen KGs, and UltraQuery, the first model capable of doing that. Directly applying an Ultra model pre-trained on one-hop link prediction does not really work - the 1p query performance would be high but it is much worse on complex queries. This happens due to the multi-source propagation issue (described in Section 4.1), that is, labeling only one node is not suitable for CLQA where at intermediate stages there might be many non-zero initialized nodes representing intermediate answers. The issue can be alleviated with scores thresholding (underperforming option) or short but effective fine-tuning on a few complex queries (better performing option). Besides, we proposed 11 new inductive CLQA datasets.
>
> > **Q1.** I am wondering if UltraQuery can exploit the feature from entity (relation) like attributes, types, and so on to improve the performance by fine-tuning.
>
> Yes, UltraQuery can be extended to support additional node / edge features by concatenating the features with zero/one vectors adhering to the labeling strategy in order to retain the expressive power of conditional MPNNs. However, there is a caveat when including features into the pre-trained model - since the model dimension is fixed, those features have to be homogeneous (having the same feature dimension and coming from the same distribution) while arbitrary datasets that might have textual / numerical / categorical features. So far, this feature alignment is quite non-trivial and there is no common approach for such an alignment in the literature, so we deem it an interesting avenue for the future work and will include this discussion into the relevant section.
>
> Having said that, the hardest, most fundamental, and widely studied setting in inductive KG reasoning [1,2,3,4,5,6, and others] is generalization without features where models have to leverage the graph structure and graph inductive biases - this is what we focus on in UltraQuery.
>
> > **Q2.** Will you present the results of the setting provided in your further work?
>
> We will do our best to provide initial results on some of the outlined problems in the final version. Meanwhile, we performed an experiment measuring the inductive performance as a function of several datasets in the training mixture. Please find the results in the attached PDF and in the general response.
>
> References:
> [1] Teru et al. Inductive Relation Prediction by Subgraph Reasoning. ICML 2020.
> [2] Zhu et al. Neural Bellman-Ford Networks: A General Graph Neural Network Framework for Link Prediction. NeurIPS 2021.
> [3] Zhang, Yao. Knowledge Graph Reasoning with Relational Digraph. WebConf 2022.
> [4] Zhu et al. A*net: A scalable path-based reasoning approach for knowledge graphs. NeurIPS 2023
> [5] Zhang et al. AdaProp: Learning adaptive propagation for graph neural network based knowledge graph reasoning. KDD 2023
> [6] Galkin et al. Towards foundation models for knowledge graph reasoning. ICLR 2024.

---

> > ### Comment · Reviewer_TgMu · 2024-08-13
> > **The response**
> >
> > Thank you for your response! I don't have any concern.

---

### Official Review · Reviewer_rSrG · 2024-07-10

**Soundness:** 4
**Presentation:** 3
**Contribution:** 3
**Rating:** 7
**Confidence:** 4

**Summary:**

This paper proposes a new framework for the generalization of complex logical question answering (CLQA). Specifically, the authors handle an extreme situation where the knowledge graph at the test time is completely different from the training time, which requires the model to adapt well to new entities and relations. The authors implement a foundation model that can construct a relation graph for any type of relation for execution. The authors examine the foundation model on generalization between extremely different knowledge graphs, which shows advantages when understanding new relations is important.

**Strengths:**

This paper establishes solid work for a unified CLQA foundation model that has strong adaptability to completely new knowledge graphs. I feel this to be an important work that benefits the training of CLQA models under low-resource situations and continuously updating knowledge graphs. The authors also widely conduct experiments to support their claims which show a strong generalization performance.

**Weaknesses:**

While I am optimistic about accepting this paper, I find it hard to get the novelty of the proposed method, whose main difference from the previous ones is the ability to handle new relations. However, the contribution on this part is not clearly presented. I feel the main problem is the lack of a detailed description of the ULTRA method and how it is adapted to the foundation model. Also, the contribution beyond the ULTRA framework needs further explanation. Still, I am glad to accept this paper for its contribution of a generally generalizable CLQA model.

**Questions:**

Please refer to the weakness part

**Limitations:**

Yes

---

> ### Author Rebuttal · Authors · 2024-08-06
>
> We thank the reviewer for appreciating our work and would like to comment on the weaknesses:
>
> > **W1.** I feel the main problem is the lack of a detailed description of the ULTRA method and how it is adapted to the foundation model.
>
> In order to build a model that zero-shot generalizes to CLQA tasks on any unseen KG we need two main components that do not depend on input entity/relation vocabulary: (1) generalizable logical operators; (2) generalizable relation projection operator. (1) is achieved by using non-parametric fuzzy logics (product logic in our case as being the most stable for gradient-based optimization). (2) is harder and requires an inductive link predictor / KG completion operator that generalizes to any graph. We integrated ULTRA as the projection operator that outputs a single scalar for each node (which is then used by fuzzy operators) that indicates a probability of this node being the answer to the query $(h, q, ?)$.
>
> Elaborated in Section 4.1, practically, ULTRA consists of two GNNs – operating on the relation level and on the entity level. First, given a query $(h, q, ?)$, ULTRA builds a graph of relations from the original KG - this graph is small and only has O(|R|) nodes and 4 specific meta-relations. Note that $|R| << |E|$ in any KG so processing this graph introduces a rather marginal computational overhead compared to the entity-level GNN. Since we know the query relation $q$, we label it with the all-ones vector in the graph of relations, run a GNN, and read all final representations as _conditionally_ _dependent_ on the starting query $q$. Those representations are used as initial edge type features in the second, entity-level GNN. Labeling the starting head node $h$ with the vector corresponding to $q$ and running a GNN over the main graph, we read out final node states as probabilities of each node being the answer to the query $(h, q, ?)$.
>
> The only learnable parameters in ULTRA are 4 meta-relations for the graph of relations and GNN weights. The 4 meta-relations represent structural patterns like “a tail of and edge with relation X is the head of another edge with relation Y” and can be mined from any multi-relational KG independent of their entity/relation vocabulary. GNN weights are optimized during pre-training. Since the model does not rely on any KG-specific entity or relation vocabulary, a single pre-trained ULTRA model can be used for zero-shot inference on any KG.
>
> Applying ULTRA for CLQA in this work, we found the multi-source propagation issue (Section 4.1), that is, ULTRA pre-trained with a single starting node, is not suitable for CLQA where at intermediate stages there might be many non-zero initialized nodes representing intermediate answers. The issue can be alleviated with scores thresholding (underperforming option) or short but effective fine-tuning on a few complex queries (better performing option). Finally, we proposed 11 new inductive CLQA datasets.
>
> Thanks for highlighting the lack of a description – we will include this more detailed discussion into the main body of the manuscript in the final version which allows one more extra page of content.

---

> > ### Comment · Reviewer_rSrG · 2024-08-07
> > **Response to Authors**
> >
> > Thanks for providing more details about the novelty and difference with important previous works. From your clarification, I get how your framework and implementation are novel against baselines. A possible way is to add which component in your framework addresses which issue in Table 1. Please consider adding these details to the main content of the paper.
> >
> > Based on the rebuttal from the authors and opinions from other reviewers, I am glad to improve my confidence and score for a **clear acceptance** since no major issue is mentioned. I appreciate the authors' effort in establishing a solid work and defending it during the rebuttal period.

---

### Official Review · Reviewer_vHv9 · 2024-07-11

**Soundness:** 2
**Presentation:** 3
**Contribution:** 2
**Rating:** 5
**Confidence:** 4

**Summary:**

The paper presents ULTRAQUERY, a groundbreaking model for zero-shot logical query answering on knowledge graphs. It introduces a novel approach that combines inductive reasoning with non-parametric fuzzy logics to generalize to new entities and relations without additional training. The model demonstrates competitive performance across various datasets and query types, setting new benchmarks in the field.

**Strengths:**

1. The paper introduces ULTRAQUERY, a novel foundation model for zero-shot logical query answering on knowledge graphs (KGs), which is a significant advancement in the field of complex logical query answering.
2. One of the key strengths of ULTRAQUERY is its ability to generalize to new entities and relations in any KG without the need for retraining, which addresses a major limitation of existing CLQA methods.
3. The paper provides extensive experimental results, demonstrating that ULTRAQUERY outperforms existing baselines on multiple datasets, showcasing its effectiveness across various query types.

**Weaknesses:**

1. While the model's generalizability is a strength, the complexity introduced by the inductive reasoning might make it challenging to scale or adapt to very large KGs.
2. The paper does not present formal theoretical results or proofs to support the empirical findings, which could have strengthened the contribution.
3. The reliance on a pre-trained model could lead to overfitting on the training dataset, potentially affecting the model's performance on unseen data. The paper could have provided more details on the computational efficiency of ULTRAQUERY, especially in the context of large-scale KGs.

**Questions:**

1. How does ULTRAQUERY handle extremely large KGs, and what are the scalability challenges?
2. Can the authors provide more insights into the decision-making process behind the choice of fuzzy logics for logical operations?

**Limitations:**

Applications to more general scenarios.

---

> ### Author Rebuttal · Authors · 2024-08-06
>
> We would like to thank the reviewer for the constructive feedback, please find our comments below.
>
> > **W1.** While the model's generalizability is a strength, the complexity introduced by the inductive reasoning might make it challenging to scale or adapt to very large KGs. The paper could have provided more details on the computational efficiency
>
> > **Q1.** How does ULTRAQUERY handle extremely large KGs, and what are the scalability challenges?
>
> Given that there haven’t been any inductive models for CLQA, we first focused on the general problem of making zero-shot inductive transfer possible. Scalability, although being somewhat orthogonal to the inductive inference and generalization problems, is a nice bonus for inductive models once inductiveness is achieved and we believe it is an encouraging direction for future work.
>
> Having said that, compared to the transductive baseline GNN-QE, the complexity overhead induced by the fully-inductive relation projection operator is rather negligible. The fully-inductive relation projection operator runs over the graph of relations and is O(|R|) where |R| is the number of unique relations. Usually |R| is orders of magnitude smaller than the number of nodes, eg, |R|=474 in FB15k237 with 15k nodes, or |R|=400 in NELL995 with 63k nodes.
>
> The support for very large KGs could be further improved by adopting more scalable entity-level GNN predictors like A*Net [1] or AdaProp [2] which have been shown to scale to graphs of millions of nodes. We are optimistic that UltraQuery could scale to such graphs as well when integrated with those models. We deem it as the next engineering step for future work and will include this discussion into the revised version.
>
> > **W2.** The paper does not present formal theoretical results or proofs to support the empirical findings, which could have strengthened the contribution
>
> We do not see why it is necessarily a weakness. Our model coincides with the most recent theoretical results in the relational GNN expressivity literature, eg, relational WL [3] and its extension to conditional MPNNs [4] paved the way for generalizable inductive models which are provably more powerful than shallow transductive embedding approaches. The product fuzzy logic we use is identified as the most stable fuzzy logic for backprop in the differentiable fuzzy logic literature [5].
> Finally, theoretical expressivity of the inductive projection operator (Ultra) is still an open question and we are not aware of the formal theoretical results about it in the literature. We believe this an intriguing question for the future work.
>
> > **W3.** The reliance on a pre-trained model could lead to overfitting on the training dataset, potentially affecting the model's performance on unseen data.
>
> Could you please elaborate on your understanding of overfitting in this case? We do not observe strong signs of overfitting on the training dataset as UltraQuery generalizes in the zero-shot manner to 20+ unseen graphs (and queries over those graphs) often better than tailored transductive models trained specifically on each target graph. In that sense, transductive models are extreme cases of overfitting as they are hardcoded to a particular set of entities or relations in a specific KG and cannot generalize to unseen graphs at inference time.
>
> What we do observe is the multi-source propagation issue (explained in Section 4.1), which might be understood as a kind of overfitting on 1p queries when using pre-trained KG completion model for CLQA. We have discussed this issue and proposed two solutions: scores thresholding and short fine-tuning.
>
> Generally, we would expect that a better pre-trained relation projection operator (with better zero-shot generalization capabilities on one-hop link prediction) would likely yield a higher performance in the UltraQuery framework as well (after doctoring the multi-source propagation issue).
>
> > **Q2.** Can the authors provide more insights into the decision-making process behind the choice of fuzzy logics for logical operations?
>
> Good differentiable fuzzy logics must be stable in the gradient-based optimization setup (like backpropagation) and do not vanish the gradient. Van Krieken et al [5] have shown that only product logic satisfies those requirements whereas other options like Gödel and Lukasiewicz t-norms and t-conorms suffer from the vanishing gradient problem. We included the main motivation in Section 4.2 and will elaborate on that in the revised version.
>
> References:
> [1] Zhu et al. A*net: A scalable path-based reasoning approach for knowledge graphs. NeurIPS 2023
> [2] Zhang et al. AdaProp: Learning adaptive propagation for graph neural network based knowledge graph reasoning. KDD 2023
> [3] Barcelo et al. Weisfeiler and Leman go relational. LoG 2022
> [4] Huang et al. A theory of link prediction via relational Weisfeiler-Leman on knowledge graphs. NeurIPS 2023
> [5] van Krieken et al. Analyzing differentiable fuzzy logic operators. Artificial Intelligence, 302, 2022

---

> > ### Author Response · Authors · 2024-08-13
> >
> > Dear Reviewer vHv9,
> >
> > Thank you for the comments and suggestions for improving our paper. As the rebuttal deadline is approaching, we would like to know whether our response addressed your concerns. We are happy to elaborate on any remaining questions.

---

### Official Review · Reviewer_5vKU · 2024-07-12

**Soundness:** 4
**Presentation:** 4
**Contribution:** 4
**Rating:** 6
**Confidence:** 3

**Summary:**

This paper proposes ULTRA QUERY, a foundation model for zero-shot logical query reasoning on knowledge graphs (KGs). Existing complex logical query answering (CLQA) methods are either transductive or only partially inductive, requiring training on each specific graph. ULTRA QUERY overcomes this limitation by deriving projection and logical operations as vocabulary-independent functions that generalize to new entities and relations in any KG. The model is pre-trained on a simple KG completion task and then fine-tuned on a single complex query dataset, enabling it to perform CLQA on unseen KGs in a zero-shot manner. Experiments show that ULTRA QUERY achieves competitive or better performance compared to baselines trained specifically on each dataset, setting a new state of the art on 15 out of 23 tested datasets.

**Strengths:**

1. ULTRA QUERY is the first foundation model for inductive CLQA that can zero-shot generalize to any KG with new entities and relations. This is a significant breakthrough as existing CLQA methods are either transductive or only partially inductive.
2. The model design is well thought out, combining an inductive projection operator based on ULTRA with non-parametric fuzzy logical operators. The fine-tuning approach on a single dataset is effective in adapting the pre-trained projection operator to the multi-hop query answering framework.
3. The ability to perform CLQA on any KG without retraining is highly valuable, potentially saving significant compute resources and time. The performance of ULTRA QUERY on various datasets demonstrates its effectiveness and practical impact.

**Weaknesses:**

1. While the experiments cover a range of datasets, a more in-depth analysis of the model's behavior under different conditions (e.g., varying graph sizes, query complexity) could provide further insights.
2. The paper does not discuss the model's scalability to very large KGs, which is an important aspect for practical applications.
3. And the scalability of the model (parameters and performance) can be further discussed.

**Questions:**

How the model scales as the pretraining data scales up?

**Limitations:**

The authors have made an effort to discuss the limitations of their work, particularly regarding the model's reliance on the pre-trained inductive KG completion model and the multi-source propagation issue.

---

> ### Author Rebuttal · Authors · 2024-08-06
>
> Thank you for appreciating our work and helpful comments.
>
> > **W1.** A more in-depth analysis of the model's behavior under different conditions (e.g., varying graph sizes, query complexity) could provide further insights
>
> We would like to point your attention to Section 5.3 in the main paper and Appendix C. Appendix C provides breakdown of the zero-shot performance w.r.t. different query types and different graph sizes. Section 5.3 includes a qualitative analysis on faithfulness as a function of varying graph sizes, and cardinality prediction.
>
> In Section 5.3 we study faithfulness (the ability to recover easy answers achievable by graph traversal, visualized as a function of increasing unseen inference graph size from inductive FB datasets) and cardinality prediction (correlation between model outputs and the true number of query answers) and show that UltraQuery performs quite competitively compared to larger transductive SOTA models while being inductive and orders of magnitude smaller.
>
> In Appendix C we plot per-query-type performance on all 14 query types across 9 increasingly larger inference graphs - showing that UltraQuery performance does not degrade on more complex patterns (such as 3p or inp) when an unseen inference graph becomes larger. In contrast, the performance of baseline GNN-QE degrades quite significantly.
>
> > **W2.** The paper does not discuss the model's scalability to very large KGs, which is an important aspect for practical applications.
>
> Thanks for bringing this question up. We will include this discussion into the future work. In this work, we first focused on the general ability to perform CLQA on any KG and scalability is the natural next step. To this end, any scalable path-based GNN like A*Net [1] or AdaProp [2] could be used as an entity-level GNN in the relation projection operator thus making it more of an engineering task. Since those GNNs can scale to graphs with millions of nodes, we are optimistic that UltraQuery could scale to such sizes as well.
>
> > **W3.** Scalability of the model (parameters and performance) can be further discussed.
>
> When training from scratch, we found marginal performance improvements beyond a hidden dimension of 64. When using a pre-trained relation projection checkpoint, we attribute most of the performance to the baseline performance of the checkpoint - since available checkpoints of Ultra are of the same parameter size, we only used the available one.
> We note that most models on knowledge graphs are designed for a single size, including the checkpoint of Ultra we use. There is no consensus in the community on what should be scaled.
>
> > **Q1.** How the model scales as the pretraining data scales up?
>
> We ran new experiments training UltraQuery on 2 (FB15k237, NELL995) and 3 (FB15k, NELL995, FB15k) CLQA datasets and measuring inductive inference performance on the rest 20 inductive datasets. Please find the charts and discussion in the attached PDF document and in the general response.
>
> References:
> [1] Zhu et al. A*net: A scalable path-based reasoning approach for knowledge graphs. NeurIPS 2023
> [2] Zhang et al. AdaProp: Learning adaptive propagation for graph neural network based knowledge graph reasoning. KDD 2023

---

> > ### Author Response · Authors · 2024-08-13
> >
> > Dear Reviewer 5vKU,
> >
> > Thank you for the comments and suggestions for improving our paper. As the rebuttal deadline is approaching, we would like to ask whether our response and new experimental results addressed your concerns. We are happy to elaborate on any remaining questions.

---

### Author Rebuttal · Authors · 2024-08-06

We thank the reviewers for appreciating our work and providing valuable feedback. We are delighted to see the work recognized as *“a significant breakthrough”*, *“a significant advancement in the field”* (**5vKU**), *“a groundbreaking model”* (**vHv9**), *“solid and important work”* (**rSrG**) with extensive experimental results (highlighted by all reviewers).

In this general response, we would like to address the common comments and report the requested experiments in the uploaded PDF:

**Model’s scalability to very large graphs (5vKU, vHv9)**

In this work, we first focused on the general ability to perform CLQA on any unseen KG and scalability is the natural next step (although being somewhat orthogonal to the inductive inference and generalization challenges). To this end, any scalable path-based GNN like A*Net [1] or AdaProp [2] could be used as an entity-level GNN in the relation projection operator thus making it more of an engineering task. Since those GNNs can scale to graphs with millions of nodes, we are optimistic that UltraQuery could scale to such sizes as well. We leave this exploration for future work and will include this discussion in the final version.

**(New experiment) Performance of UltraQuery trained on more datasets (5vKU)**

As per reviewer 5vKU request, we conducted a new experiment measuring the inductive inference CLQA performance depending on the number of KGs in the training mixture. Please find the details and full results in the attached PDF.

The original model (1G) was trained on queries from the FB15k237 dataset.
In order to maintain the zero-shot inductive inference setup on 11 inductive $(e,r)$ and 9 inductive $(e)$ datasets (total 20 datasets), we trained new versions on the rest of BetaE datasets: FB15k237 and NELL995 queries (2G), and FB15k273, NELL995, and FB15k queries (3G). A summary table of average EPFO MRR results is provided below:

| Model | Inductive (e,r) (11 datasets) | Indudctive (e) (9 datasets) | Total Avg (20) |
| :---: | :---: | :---: | :---: |
|  | EPFO avg MRR | EPFO avg MRR | EPFO avg MRR |
| UltraQuery 1G | 0.280 | 0.312 | 0.296 |
| UltraQuery 2G | 0.310 | 0.307 | 0.308 |
| UltraQuery 3G | 0.304 | 0.292 | 0.298 |

Generally, training on more graphs increases the performance on EPFO queries on 11 inductive $(e,r)$ datasets and slightly improves the overall performance on 20 datasets. Having said that, we identify several factors that might affect the performance:
* Given a short time for running experiments, the training configuration might be suboptimal and might require more hyperparameter tuning including the sampling ratio from each dataset or adversarial temperature for the loss function;
* Newly included datasets, NELL995 and FB15k, might not be very useful for inductive datasets which are based mostly on Freebase and Wikidata. That is, NELL995 is sparse and combines several disconnected components whereas FB15k is known for test set leakages. We hypothesize that adding a few datasets from inductive $(e)$ and $(e,r)$ benchmarks to the training mixture might yield much better zero-shot performance. We leave the study of the most effective data mixtures for the future work;

We will include this experiment in the final version as well. We would be happy to engage in the discussion in the comments, please let us know if you have any further concerns.

References:
[1] Zhu et al. A*net: A scalable path-based reasoning approach for knowledge graphs. NeurIPS 2023
[2] Zhang et al. AdaProp: Learning adaptive propagation for graph neural network based knowledge graph reasoning. KDD 2023.

---

### Decision · Program_Chairs · 2024-09-25

**Decision:**

Accept (poster)

**Comment:**

This work is built on top of the transferable graph representations from (ULTRA[15]), and fuzzy logics. In order to achieve multi-step reasoning it employees
1) inductive parametric projection operators (IPP, similar to those of ULTRA) to produce a scalar score for each node in KG given starting or intermediate node information.
2) non-parametric logical operators (fuzzy logics) to perform logic operation over sets of weights.

The projection operator is fine-tuned from the ULTRA model's project operations (cross entropy loss on the final answer) to bridge the gap between link prediction and CLQA.  Experiment on 23 datasets shows comparable (yet lower) performance on inductive (e,r) tasks, compared to Inductive(e) and transductive tasks.

All reviewers recognize the importance of the problem and the demonstrated capability. However, they also have concerns about the lack of clarity for the novelty and theoretical contribution, and the scalability with large KGs. The paper can benefit from more detailed analysis and explanation of the projection/logical operators and computational complexities.